# An unexpected *N*-dependence in the viscosity reduction in all-polymer nanocomposite

Tao Chen[1,3], Huan-Yu Zhao [1,3], Rui Shi [1,3], Wen-Feng Lin[1], Xiang-Meng Jia[1], Hu-Jun Qian[1]*, Zhong-Yuan Lu[1], Xing-Xing Zhang[2], Yan-Kai Li[2] & Zhao-Yan Sun[2]

Adding small nanoparticles (NPs) into polymer melt can lead to a non-Einstein-like decrease in viscosity. However, the underlying mechanism remains a long-standing unsolved puzzle. Here, for an all-polymer nanocomposite formed by linear polystyrene (PS) chains and PS single-chain nanoparticles (SCNPs), we perform large-scale molecular dynamics simulations and experimental rheology measurements. We show that with a fixed (small) loading of the SCNP, viscosity reduction (VR) effect can be largely amplified with an increase in matrix chain length *N*, and that the system with longer polymer chains will have a larger VR. We demonstrate that such *N*-dependent VR can be attributed to the friction reduction experienced by polymer segment blobs which have similar size and interact directly with these SCNPs. A theoretical model is proposed based on the tube model. We demonstrate that it can well describe the friction reduction experienced by melt polymers and the VR effect in these composite systems.

[1] State Key Laboratory of Supramolecular Structure and Materials, Institute of Theoretical Chemistry, Jilin University, Changchun 130023, China. [2] State Key Laboratory of Polymer Physics and Chemistry, Changchun Institute of Applied Chemistry, Chinese Academy of Sciences, Changchun 130022, China. [3]These authors contributed equally: Tao Chen, Huan-Yu Zhao, Rui Shi. *email: hjqian@jlu.edu.cn

Adding nanoparticles (NPs) into polymer melts can lead to fascinating improvements in materials mechanical[1–4], optical[5–7], and electrical[8] properties. It is both fundamentally and practically important[9,10] to understand the underlying mechanisms of NPs manipulation on material properties. However, due to complex influencing factors in the system, many experimentally reported peculiar phenomena are yet far from being fully explored. For instance, early seminal works by Mackay and coworkers reported a dramatic reduction in viscosity of PS melt due to the addition of a small amount of cross-linked single-chain nanoparticles (SCNPs)[11–13]. This phenomenon is in a striking contrast to Einstein-Batchelor[14,15] law which predicts a viscosity increase by incorporating NPs in the system. It also remains so far a major unresolved puzzle in polymer field.

Earlier works can be traced back to the work of Malinskii and coworkers in the 1970s, where an anomalous reduction of the PVC melt viscosity caused by the addition of NPs was reported[16], it was attributed to the creation of additional free volume at the polymer/NP boundary surface region[16,17]. However, it reverted to a viscosity increase with a further increase in NP loading ($\phi$). Such a transition from viscosity reduction (VR) to increase was confirmed recently in a poly(ethylene-alt-propylene)/silica nanocomposites[18]. A recent molecular dynamics simulation study[19] showed that with a small loading of NPs, VR effect can be attributed to the disentanglement effect found for melt polymer chains. However, NPs were found to be "entangled" when its loading reached a critical value and hence melt chain relaxations were very much hindered. Other than these reports, VR was also widely reported in the composite systems of polymers with inorganic NPs[13,20,21], organic fullerenes[13,22], grafted silica NPs[23,24], and dendrimers[25]. Various polymer and NP types in these systems would result in different types of interactions. In contrast, these complex enthalpic interactions do not exist in the all-polystyrene (PS) composites reported by Mackay and coworkers[11–13]. Therefore all-PS composite can serve as an ideal model system. In addition, the interface region in such all-PS composite system has a liquid-liquid contact between PS SCNPs and melt polymers, which is also different from a solid-liquid like interface existing in inorganic NP/polymer composites. From this point of view, a recently reported dendritic polyethylene (dPE)/PS nanocomposite[25] has the same characteristic of soft interface. More importantly, adding a very small amount of either PS SCNPs or dPE into PS melts can cause a dramatic viscosity reduction: Tuteja et al.[12] reported that the PS melt viscosity can be reduced up to 80% by adding 1% PS SCNPs, while similarly addition of 5% dPE NPs[25] in PS melt can cause a reduction of melt viscosity up to 95%. Such an abrupt decrease in viscosity is obviously beyond conventional disentanglement effect, which is predicted to be proportional to the volume fraction of the NPs[19]. On the other hand, available experiment results[12,25] showed that the plateau modulus ($G_N^0$) is nearly unaffected at low NP loadings, indicating no obvious change in the melt chain entanglements according to $G_N^0 \sim \rho_e k_B T$[26], where $\rho_e$, $k_B$, and $T$ are the number density of entanglements, Boltzmann constant and temperature, respectively. Therefore, the large reduction in viscosity in these systems cannot be simply explained by disentanglement effect. At the same time, according to Goldansaz et al.[25], such VR cannot be attributed to confinement effect, free volume effect, surface slippage, shear banding, or particle induced shear thinning.

Our previous simulations[27] with a single PS SCNP presented in the system showed that the relaxation of the surrounding melt PS chains can be accelerated in the vicinity of the SCNP. In addition, we also showed[28] that when these SCNPs contain 250 styrene monomers each and 20% of them are cross-linking units (exactly the same as we use here in this study), they interact directly with linear melt PS chains on a length sacle of the SCNP size, i.e., we found direct evidences showing that these SCNPs were dynamically coupled with polymer chain segment blobs with a similar size as these SCNPs. In order to understand the mechanism of the substantial, nonlinear VR in all-PS composite system[11,12], large-scale coarse-grained molecular dynamics simulations are performed in this study. The polymer/NP composite (PNC) systems with different chain lengths and different volume fractions of SCNPs are systematically investigated. Interestingly, we find an unexpected $N$-dependence for the acceleration in the chain relaxation dynamics. We also carry out experimental rheology measurements of the corresponding all-PS composite systems, predictions from simulations are confirmed, i.e., VR effect is found to be larger in systems with larger molecular weight for linear polymer chains. Based on both simulation and experimental results, we propose a theoretical model based on the tube model to understand the VR effect and its $N$-dependence.

## Results

**Influence of NPs on melt polymer chain conformation.** First of all, we calculate the average values of end-to-end distance $R_{ee}$ and radius of gyration $R_g$ of polymer chains in both pure melt and composites with different NP loadings, the results are listed in Supplementary Table 2. It shows that the presence of NPs has almost no effect on the chain dimensions, except a very small decrease in $R_{ee}$ and $R_g$ at very hight NP loadings. The same results had been reported in other simulations, a nice review/discussion on this topic can be found in a recent review article by Kröger and cowokers[29]. The primitive path analysis[19,30–32] using Z1 code[30–32] shows that the length of the primitive path $L_{pp}$ in PNC system even at hight SCNP loading still follows Gaussian distribution (see Supplementary Fig. 2), which is in good agreement with experiment[33] and other simulation[19] results. The average value of $\langle L_{pp} \rangle$ and its standard deviation $\sigma_{L_{pp}}$ are listed in Supplementary Table 3. Note that although $\langle L_{pp} \rangle$ is reduced at high NP loadings, the $\sigma_{L_{pp}}$ remains almost unchanged, which implies that NP has no effect on the fluctuation of $L_{pp}$. Namely, it has nearly no effect on the contour length fluctuation (CLF) of the melt chains, which is believed to be crucial to the melt chain dynamics.

**Entanglement between polymer chains in simulation.** For the pure PS720 system, the average entanglement strand length obtained from modified coil estimator[34] in primitive path analysis is $\langle N_e \rangle = 163.7$, which is in a good agreement with experimentally measured entangled molecular weight of 17 kDa, as listed in the text book by Rubinstein and Colby[26] (see in Table 9.1 on page 362). Therefore PS720 chain has ~4 entanglement strands per chain on average. In addition, we calculate the mean square displacement (MSD) of the two innermost monomers ($g_1(t)$) and that of the center of mass (CM) ($g_3(t)$) for PS720 chains. The MSD is defined as $g(t) = \langle |r(t) - r(0)|^2 \rangle$, where $r(t)$ is the coordinate of either the two innermost monomers for $g_1(t)$ or the CM of the PS chain for $g_3(t)$ at time $t$. The results are shown in Fig. 1. Although we do not plot the results at small time scale, $g_1(t)$ and $g_3(t)$ show different scaling relations in distinct regions[26,35]. For instance, $g_1(t)$ scales as $\sim t^{1/2}$ at short time scale $t < \tau_e$, followed by a lower scaling of $g_1(t) \sim t^{1/4}$ in the tube ($\tau_e < t < \tau_R$) and thereafter $g_1(t) \sim t^{1/2}$ (with $t < \tau_d$) and $g_1(t) \sim t^1$ (with $t > \tau_d$). Here $\tau_e$, $\tau_R$ and $\tau_d$ are the relaxation time of an entanglement strand, Rouse time, and disentanglement time of the polymer chain respectively. For MSD of chain CM, $g_3(t) \sim t^{1/2}$ with $t < \tau_R$ and $\sim t^1$ with $t > \tau_R$. These results indicate that PS720 chains are entangled. Note that the slope in the intermediate regime is a little larger than a true $t^{1/4}$ power since the

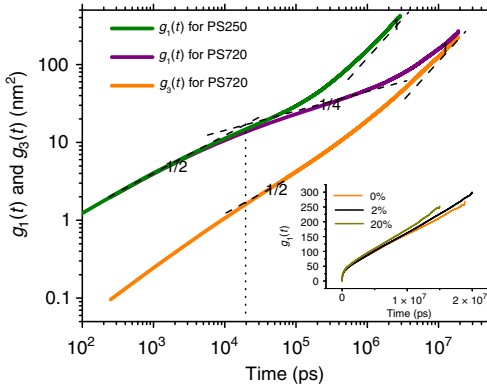

**Fig. 1** The MSD of the two inner-most monomers $g_1(t)$ on chain backbone for both PS250 and PS720 chains, and that of the center of mass (CM) $g_3(t)$ for PS720 chains. The dash lines with scaling factors (1/4, 1/2, and 1) are plotted as references. The vertical dotted line indicates an intersection at $\tau^* \approx 20$ ns in $g_1(t)$. The inset shows the comparison of $g_1(t)$ for pure PS720 melt and two composite systems with NP loading of 2% and 20%.

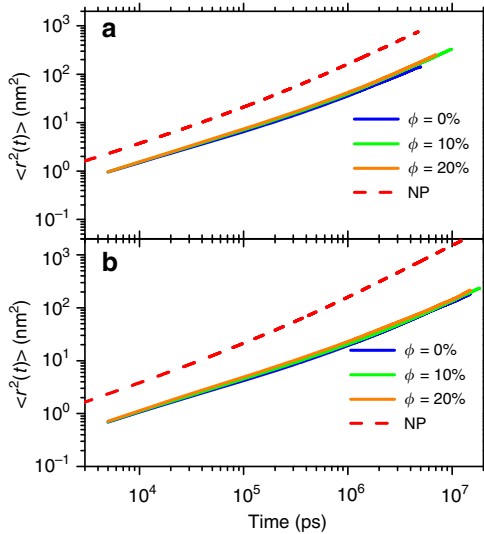

**Fig. 2** Solid lines are the MSD of linear matrix PS chains in the composites of **a** PS500/NP250 and **b** PS1300/NP250 with different loadings ($\phi$) of NP250. The MSD for the CM of NPs are plotted as references (dash line).

system only has ~4 entanglements per chain. For the monomer MSD curve, a transition is found at $\tau^* \approx 20$ ns from short time scale behavior with $g_1(t) \sim t^{1/2}$ to intermediate time scale with $g_1(t) \sim t^{1/4}$. Therefore $\tau_e = 23.2$ ns according to the relation $\tau_e = \frac{36}{\pi^3} \tau^*$ given by Likhtman and McLeish[36]. In addition, $g_1(t)$ is also calculated for PS250 and shown in Fig. 1. Although we know that entanglement in this system is very weak since each chain has $N/N_e < 2$ entanglements, a definite decrease of the slope in the intermediate regime is observed after $\tau_e$. Before $\tau_e$, it has exactly the same behavior as in PS720 system.

**Acceleration in polymer chain dynamics**. To characterize the translational dynamics of polymer chains, we calculate the MSD of the CM of polymer chains in both composite and pure polymer melt systems. The results are shown in Fig. 2. The linear PS chains in two systems are 500 or 720 styrene monomers long and indicated as PS500 and PS720 respectively. In both systems, MSDs of PS chains in composite are faster than those in pure melt. Note that the differences between results from composite

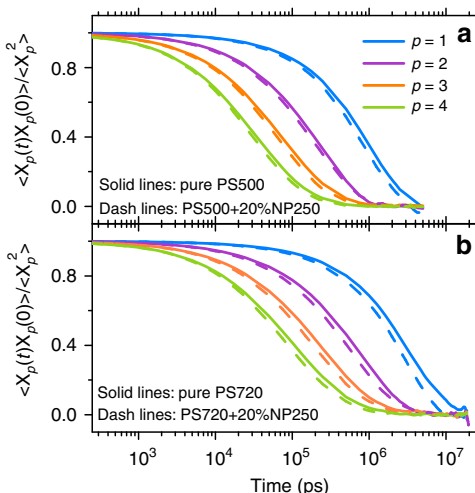

**Fig. 3** The time autocorrelation function for four lowest Rouse modes of linear PS chains in **a** PS500 and **b** PS720 systems. Solid lines are for pure PS melts and dash lines are for PNCs with a 20% loading of NP250, respectively.

systems with different NP loadings are small due to log scale we adopted. Plots of the same data with linear scale are supplied in Supplementary Fig. 3, which shows clearer differences. Short dash lines in Fig. 2 are MSD curves for NPs in representative PNC systems with a SCNP loading of 20%, which shows NPs diffuse much faster than melt chains. It is interesting to see that SCNPs seem to follow similar Rouse-like time dependent behavior as those of the CMs of the PS chains, i.e., there is a transition from a sub-diffusive behavior at short time scale to a Fickian regime at long time scale, though this transition happens a little earlier for SCNPs. We attribute this similarity to the length-scale dependent interaction between these SCNPs and the matrix polymer chains. Specifically, in our previous simulations[28,37], SCNPs are found to interact directly with chain segment blobs which have similar size as these SCNPs, and we observe that such chain segment blobs have ~144 monomers each and a little smaller size than entanglement strand. Therefore the transition from sub-diffusive to Fickian occurs a little earlier for SCNPs.

In Fig. 3 we plot the Rouse mode analyses results of $\langle \mathbf{X}_p(t) \cdot \mathbf{X}_p(0) \rangle / \langle \mathbf{X}_p^2 \rangle$ for both PS500 and PS720 pure melts and composite systems containing 20% NP250, here $\mathbf{X}_p$ is the Rouse mode as defined in Supplementary Eq. (1). Results with mode indices of $p = 1 \sim 4$ are plotted, which are the slowest relaxation modes in our simulated systems. It shows that these modes have been fully relaxed. We plot in Supplementary Fig. 3 the values of $\beta_p$ (see Supplementary Eq. (4)) as a function of $N/p$ for the first 20 modes for pure PS melts with different $N$. The minimum of $\beta_p$ is found at ~166, which is close to $N/p \approx N_e$ and is consistent with the results as reported in ref. [19] in the aspect that the minimum of $\beta_p$ appears at the entanglement length. More importantly, results in Fig. 3 show that the relaxation of chains in the composite system (dashed color lines) are much faster than those in pure melts (black lines).

**Unexpected $N$-dependence in the acceleration of chain dynamics**. From the Rouse mode analysis, we obtain the relaxation times $\tau_p^{\text{eff}}$ at different $p$ modes, which correspond to the effective relaxation time of melt chain segments with $N/p$ monomers, details can be found in Supplementary Eq. (4). The results are plotted in Fig. 4a for systems with different chain lengths but a fixed NP loading of 20% and the results are

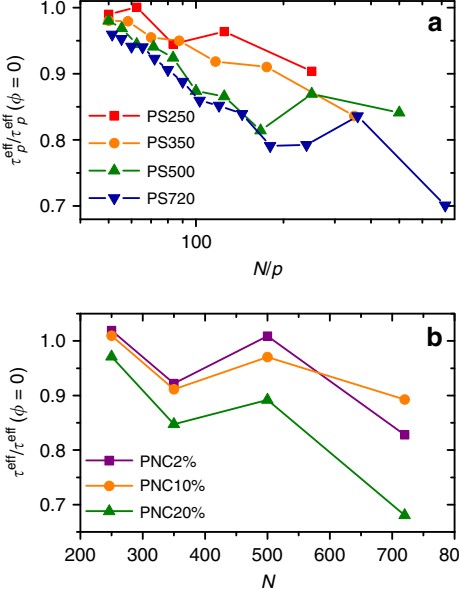

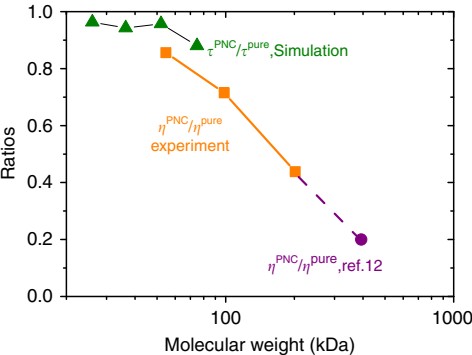

**Fig. 5** The relative viscosity $\eta^{PNC}/\eta^{pure}$ for the systems with different PS molecular weight at a fixed NP loading of 2% at T = 170 °C, $\eta^{PNC}$ and $\eta^{pure}$ are the viscosities of PNC and pure PS melt systems respectively. NP has a molecular weight of 26.2 kDa. The last data point in purple is taken from ref. [12], where NP loading is 1% and SCNP has a molecular weight of 25 kDa. Triangles are the ratio between effective terminal relaxation times for composite systems with a SCNP loading of 2% and that of the pure melt, this ratio is taken from Fig. 4a at p = 1.

**Fig. 4** Effective relaxation time ratio between composite and neat system. **a** The $(N/p)$-dependent segmental effective relaxation time $\tau_p^{eff}$ obtained from Rouse mode analyses is plotted for matrix chain segments with an effective length of $N/p$ in composite systems containing 20% NP250, only values with $N/p > 50$ are plotted; **b** the $N$-dependent chain effective relaxation time $\tau^{eff}$ obtained from $<R(t)R(0)>$, $R(t)$ is the end-to-end vector of the melt PS chain at time $t$. $\tau^{eff}$ in both figures are normalized by corresponding values from pure melt systems $\tau^{eff}(\phi = 0)$.

normalized by the corresponding $\tau^{eff}(\phi = 0)$ values obtained in pure PS melts. Interestingly, we find an obvious chain length $N$-dependent cascaded acceleration in chain relaxation. Overall, the ratio $\tau_p^{eff}/\tau_p^{eff}(\phi = 0)$ decreases with length of chain segment. Such $N$-dependent cascaded acceleration effect is twofold: (i) As indicated by data points with mode index $p = 1$, there is apparently larger acceleration in chain relaxation for longer chains. Similarly for the segments with length $N/p$ in a specific system, longer ones will have larger acceleration since the ratio $\tau_p^{eff}/\tau_p^{eff}(\phi = 0)$ decreases with $N/p$ in all systems. (ii) For segments with the same length $N/p$ but in systems with different total chain length $N$, acceleration in its relaxation also has an $N$-dependence, larger acceleration can be found in system with larger $N$. For instance, all data points for the system PS720 with the longest chain length are overall located at the bottom among all systems. Note that this phenomenon still holds if we calculate the effective relation time from $<R(t)R(0)>$ (see Supplementary Eqs. (5) and (6)), the results are shown in Fig. 4b. Encouragingly, for the same SCNP loading of 20%, relaxation time ratios are found very similar for two algorithms, i.e., values for the last points of each system in Fig. 4a with $p = 1$ are very similar to the values in Fig. 4b (triangles).

**Rheology measurements**. In order to testify the predictions from the simulation, experimental rheology (small amplitude oscillatory shear) measurements are performed. The composite systems are composed of linear polystyrene chains with molecular weights of 54.6, 98.5, and 201.8 kDa, and PS SCNPs with a molecular weight of Mw = 26.2 kDa. Linear polystyrenes were purchased from Polymer Source Inc., they were purified via a reprecipitation process before use. PS SCNPs were synthesized following the procedure proposed by Hawker et al.[38]. Details of the synthesis and characterization procedures for PS SCNPs, preparation of the composites, system details and rheology measurements are

supplied in the section of Experimental details in Supplementary Information.

The first result we look at is the scaling of zero-shear visocity with respect to $N$ in pure polymer melt without any NPs, the results are plotted in Supplementary Fig. 4, which shows a scaling of $\eta \sim N^{3.44}$ at 170 °C. Our data also show very good agreement with earlier measurements[12,39]. After adding 2% NP in the system, the viscosity of the composite system is dramatically reduced. Figure 5 shows the ratio between the zero-shear viscosities of composite system and pure PS melt at 170 °C, $\eta^{PNC}/\eta^{pure}$, data are listed in Supplementary Table 7. It shows that the ratio $\eta^{PNC}/\eta^{pure}$ has a strong $N$ dependence. At a fixed NP loading, composite systems with longer chains will have larger VR. For the system with a small molecular weight of 55 kDa, the viscosity of the composite system is moderately reduced, it has a viscosity ratio of $\eta^{PNC}/\eta^{pure} = 0.86$, while for the system with a high molecular weight of 202 kDa, the viscosity ratio is largely reduced to 0.44. This ratio can be further decreased to 0.20 when the molecular weight of melt PS is 393 kDa as reported by Tuteja et al.[12] at a NP loading of 1%, as indicated by a solid circle in Fig. 5. These results are in a good agreement with $N$-dependent reduction in $\tau_{eff}$ as reported in Fig. 4. For comparison, the relaxation time ratios ($\tau_{PNC}/\tau_{PS}$) from simulation are also plotted in Fig. 5 for the composite with a SCNP loading of 2% (triangles in the figure). Agreement between simulation and experiment is encouragingly good. Supplementary Fig. 8 shows the representative results of the complex shear viscosity as a function of frequency for both pure melt with a molecular weight of 202 kDa for linear PS chains and its composite with an NP loading of 2%. It shows a dramatic decrease of shear viscosity in Newtonian regime.

**Disentanglement effect**. The number of entanglements per chain $\langle Z_{kink} \rangle$ obtained from Z1 analysis is plotted in Fig. 6 versus relaxation time ($\tau^{eff}$) of the melt chains at various NP loadings, the corresponding data are listed in Supplementary Table 4. Both values of $\langle Z_{kink} \rangle$ and $\tau_{eff}$ are normalized by the corresponding values obtained in pure melt systems at $\phi = 0$. Note that Rouse mode analysis[19,40–42] (see Supplementary Eqs. (1–4)) is performed to calculate the relaxation time $\tau^{eff}(p)$ at different $p$ modes, from which $\tau^{eff}(p = 1)$ is taken as the chain relaxation time $\tau^{eff}$. Upon increase of NP loading ($\phi$) in the system, we do

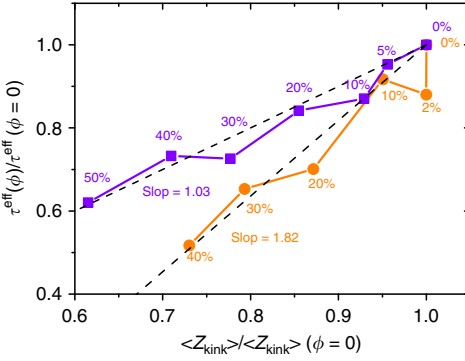

**Fig. 6** Chain relaxation time $\tau^{eff}$ at $p = 1$ plotted as a function of average number of entanglements per chain $\langle Z_{kink}\rangle$ in composite systems. 'PS500/NP250' (squares) and 'PS720/NP250' (circles) with different NP loadings ($\phi$). They are both normalized by the corresponding values in pure PS melts with $\phi = 0$.

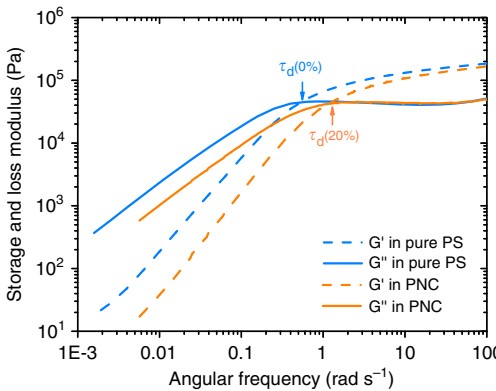

**Fig. 7** Experimentally measured storage ($G'$) and loss ($G''$) modulus for both pure melt and composite system. Melt chains in both systems have a molecular weight of 202 kDa, composite contains 2% of NPs with a molecular weight of 25 kDa.

find a disentanglement effect in the system, i.e., $\langle Z_{kink}\rangle$ value decreases with the addition of NPs. At the same time, a decrease is also found in the chain relaxation time $\tau_d$. However, a stronger dependence of $\tau^{eff}$ over $\langle Z_{kink}\rangle$ is found in composite system with $N = 720$ ('PS720/NP250') than that with $N = 500$ ('PS500/NP250'). Least squares fittings result in slopes of 1.03 and 1.82 for systems PS500/NP250 and PS720/NP250 respectively, as shown in Fig. 6. Intuitively, disentanglement effect caused by the presence of NPs could be a direct possible reason for the acceleration in the chain dynamics. According to the tube model, chain relaxation time $\tau^{eff} \sim \tau_R \left(\frac{N}{N_e}\right) \sim \tau_R Z$, where $\tau_R$ is the Rouse time of the chain and $Z = \frac{N}{N_e}$ is the chain entanglement density. Therefore, a simple relation $\langle Z_{kink}\rangle/\langle Z_{kink}^0\rangle = \tau^{eff}(\phi)/\tau^{eff}(\phi = 0)$ can be derived since the addition of the NPs in the system has almost no influences on the chain dimension of the melt polymers (see listed $R_{ee}$ and $R_g$ values in Supplementary Table 2 for melt polymer chains), a similar use and derivation of this relation can be found in ref. [19]. Although this relation is valid for composite system 'PS500/NP250' (squares in Fig. 6), a stronger dependence is found in composite system 'PS720/NP250' (circles in Fig. 6) where the reduction in the relaxation time is found much faster than the reduction in entanglements. For instance, a NP loading of $\phi = 20\%$ can only cause a reduction of ~10% in $\langle Z_{kink}\rangle$ but a much larger reduction of ~30% in $\tau^{eff}$ is found. Note that a linear fitting in Fig. 6 for system 'PS720/NP250' is only guided to the eye, we do not claim a rigorous use of this relation. In addition, we have also simulated a PS720 nanocomposite system which contains only 2% NPs. The relaxation time ratio is found to be $\tau^{eff}/\tau^{eff}(\phi = 0) = 88\%$, while the $\langle Z_{kink}\rangle$ remains unchanged. According to these results, we conclude that disentanglement effect cannot fully account for the acceleration in chain dynamics found in the system. Here we have to note that according to Li and coworkers[19], adding solid NPs can cause confinement effects at large NP loadings. We do not observe such confinement effects even at very high loadings of SCNPs, this can be attributed to the inherent softness of these SCNPs.

Experimentally, we plot the measured results of storage and loss modulus ($G'$ and $G''$) for both pure melt and composite systems in Fig. 7. Here the melt chain has a molecular weight of 202 kDa and the composite has an NP loading of 2%. For pure polymer melt system, the entanglement molecular weight is $M_e \approx 16.38$ kDa according to $G_0 = \frac{4\rho RT}{5M_e}$, this value is very close to a previously reported experimental value of $M_e = 17$ kDa[26]. In addition, the value of $M_e \approx 16.38$ kDa is in a good agreement

with $N_e = 163.7$ obtained from Z1 calculation in our simulation. More importantly, we see that there is a large right shift of the point for $G' = G''$ for the composite system with respect to that of pure melt, which indicates a large reduction in disentanglement time, which is consistent with the observed VR. However, the composite system has an almost identical plateau modulus ($G_0$) as pure melt system. It indicates that the two systems have almost identical entanglement density in the system. This result confirms that the mechanism of disentanglement effect is not the one responsible for the VR.

**Molecular mechanism.** To the best of our knowledge, such $N$-dependent acceleration in entangled polymer chain dynamics has not been reported before. The results in the above section show that the disentanglement effect cannot be used to explain the abrupt reduction in viscosity. To explore the underlying molecular mechanism, a full understanding of the influence by SCNP on surrounding chain dynamics is crucial. Our previous simulations[27] have demonstrated that such internally cross-linked SCNPs are soft in nature, and the dynamics of surrounding polymer chain segments are largely accelerated. Such acceleration effect at the SCNP surface area can be understood since the SCNPs are identical to polymer chains in its composition but internally cross-linked, therefore the internal degrees of freedom are largely reduced for these SCNPs, which will result in a reduction in friction strength felt by surrounding polymer chains, consequently a speed-up of surrounding chain dynamics can be expected. In addition, interpenetration/contacts between surrounding melt chain segments in the interior of the SCNP will be much reduced comparing with those between free melt polymer chains. To provide a direct evidence, we calculate the MSD of the two inner most monomers on melt chain PS720 in both composite and pure melt systems, results are shown in the inset of Fig. 1. It clearly shows a speed-up effect in the composite system. Such an acceleration effect directly indicate that frictions from SCNPs on surrounding melt chain monomers are reduced comparing with frictions between free polymer chain monomers in pure melt, note that the composite systems have almost the same density as pure melt.

On the other hand, the SCNP has generally a faster diffusion than the melt chain since the SCNP has an overall spherical shape on average and a smaller molecular weight. Representative MSD curves of NP250 in PS500 and PS720 melts are plotted as dash lines in Fig. 2, SCNPs are found to diffuse much faster than the melt chains. In addition, recent theoretical works[43,44] predicted that the NP will have a constant diffusion coefficient in long chain

melts where the NP is smaller than both radius of gyration and tube diameter of the melt chain, which is also verified in our recent simulations[37] for the current all-PS composite system. It can be attributed to the fact that when SCNP diffuses in polymer melt, it only feels the friction from the surrounding polymer chain segments which have similar size with SCNP. Therefore, when the melt chain length increases at a fixed loading of the SCNPs, the diffusion coefficient of SCNP does not change although that of the chain decreases cubically with $N$ according to the tube model. These results imply that interactions between SCNP and melt polymers are on the length scale of the SCNP size. Our previous simulations[28] also provide direct evidences for a dynamic coupling between these SCNPs and melt polymer chains on such a length scale, i.e., for polymer chain segments having a similar size with SCNP. Note that these segments contain ~144 styrene monomers each, which is smaller than but similar with an entanglement strand, $N_e$, of melt polymers.

From the above analyses, we know that SCNP in the system interacts with melt PS chains on a length scale of SCNP size or entanglement strand. Knowing such a detailed interaction mechanism between SCNPs and free melt polymer chains will be also the key to understand the observed abnormal $N$−dependence in chain acceleration or friction reduction. The discussions in the following will be focused on the scale of entangled strands, they are also referred to as blobs in the following.

As we discussed above, interactions from SCNPs will effectively reduce the frictions and therefore accelerate the dynamics of surrounding blobs. For instance, if a blob $i$ interacts with an SCNP the friction of the blob $i$ will be reduced and therefore its dynamics will be accelerated. Such acceleration effect will not disappear immediately while the SCNP leaves, instead it will also effectively accelerate the dynamics of the neighboring blobs along the chain contour due to the chemical connectivity between these blobs. Namely chain blobs are dynamically correlated along the chain contour. Based on the above consideration, we simply use a summation in the following equation to take account of the contribution of each $j$ blob on the same chain to the variation of the friction on blob $i$ at a given SCNP fraction $\phi$,

$$\frac{d\zeta_i(\phi)}{d\phi} = \sum_{j=1}^{Z} -\delta A_{ij}\zeta_j(\phi), \quad (1)$$

$\zeta(\phi)$ is the friction constant experienced by blobs at a given SCNP loading $\phi$, it converges to $\zeta^0$ in pure polymer melt with the absence of SCNPs at $\phi = 0$. $\delta$ describes an effective reduction in $\zeta_i$, $Z = N/N_e$ is the number of entanglements per chain, $A_{ij}$ describes the correlation between blobs or the contribution from neighboring $j$ blobs along the chain backbone to blob $i$. Eq. (1) can be rewritten as,

$$\frac{d\zeta}{d\phi} = -\delta A \zeta, \quad (2)$$

with $A := A_{ij}$ and $\zeta := (\zeta_1, \zeta_2, \cdots, \zeta_Z)^T$. It has a generic solution of,

$$\zeta = \exp(-\phi\delta A)\zeta^0 \quad (3)$$

Therefore, curvilinear diffusion coefficient of the polymer chain in the tube can be written as,

$$D_c(\phi) = \frac{k_B T}{\sum_{i=1}^{Z} \zeta_i(\phi)} = \frac{k_B T}{\sum_{i=1}^{Z} \exp(-\phi\delta A)\zeta^0}. \quad (4)$$

Suppose correlation between blobs $A_{ij}$ decays exponentially on chain backbone, $A_{ij} = \exp(-\alpha|j - i|)$, where $1/\alpha$ is the correlation length between blobs. If we define the denominator

of Eq. (4) as,

$$f(\phi; Z) = \frac{1}{Z}\sum_{i=1}^{Z} \exp(-\delta\phi A) : \mathbf{1}_{Z \times Z} \quad (5)$$

This function defines the dependence of friction coefficient on $\phi$ and $Z$. As a consequence, the curvilinear diffusion coefficient in Eq. (4) will be,

$$D_c = \frac{k_B T}{Z\zeta^0 f(\phi; Z)}. \quad (6)$$

Eventually, the polymer chain relaxation time can be written as[45],

$$\tau_{rep} = \frac{L^2}{\pi^2 D_c} = 3\tau_e Z^3 f(\phi; Z), \quad (7)$$

where $L$ is the length of the primitive path of the tube, $\tau_e = N_e^2 b^2 \zeta^0 / 3\pi^2 k_B T$ is the Rouse time of the entanglement stands. After considering the term for CLF effect, it turns out to be,

$$\tau_{rep} = 3\tau_e Z^3 \left[1 - \mu\sqrt{\left(\frac{1}{Z}\right)}\right]^2 f(\phi; Z) = \tau_{rep}^{pure} f(\phi; Z). \quad (8)$$

Similarly, the viscosity can be written as,

$$\eta = \frac{\pi^2 k_B T}{4\nu_0 N_e} \tau_e Z^3 \left[1 - \mu\sqrt{\left(\frac{1}{Z}\right)}\right]^3 f(\phi; Z) = \eta^{pure} f(\phi; Z). \quad (9)$$

In the above Eqs. (8) and (9), $\tau_{rep}^{pure}$ and $\eta^{pure}$ are the original definition of chain relaxation time and viscosity for entangled chains of the tube model.

According to our analyses in the above section, a small amount of SCNP loaded in the system has almost no effect on the chain conformations, i.e., composite system has an almost identical $N_e$ value as pure system. This is also verified in the analysis in Fig. 7. Therefore we have $\eta^{PNC}(\phi)/\eta^{pure} = f(\phi; Z)$ according to Eq. (9). For the currently investigated composite systems, this ratio is plotted in Fig. 8. A least squares fitting of these data points using the relation in Eq. (9) results in parameters $\phi\delta = 0.0792$ and $\alpha = 0.0759$. It corresponds to a reasonable correlation length of $1/\alpha \sim 13$ entanglement blobs. Note that in the form of Eq. (5), $A$ is a matrix and $i = 1, 2, \cdots, Z$ is the integers indicating the number of entanglements per chain. While for polymers in our experiment, we have decimal numbers of $Z$. Therefore fitting in Fig. 8 is already quite good, although experimental data points are not exactly located on the fitting curve. A schematic depiction of such interactions between SCNP and polymer chain segment blob is drawn in Fig. 8 on the right.

With the parameters of $\phi\delta = 0.0792$ and $\alpha = 0.0759$, we plot friction coefficient ratios, $\zeta_{PNC}(ib)/\zeta_0$, for our experimental systems with diferent molecular weights in Fig. 9 using the definition in Eq. (3). $\zeta_{PNC}(in)$ and $\zeta_0$ are friction coefficient on blobs in the composite and pure PS melt respectively, $ib$ is the index of blobs along the chain contour. We see that there is a $N$-dependence in this ratio, system with larger molecular weight has a larger reduction in the friction, corresponding to a larger VR effect we found. Also it changes along the chain as we expected, termial blobs have a lower reduction in friction than the inner ones due to a dynamical decay of the acceleration effect caused by the interaction between SCNP and these blobs.

## Discussion

In this work, by performing large-scale molecular dynamics simulations and experimental rheological measurements, dynamics in an all-polymer composite system composed of linear polystyrene chains and internally cross-linked SCNPs are investigated. We demonstrate that adding soft SCNPs can dramatically

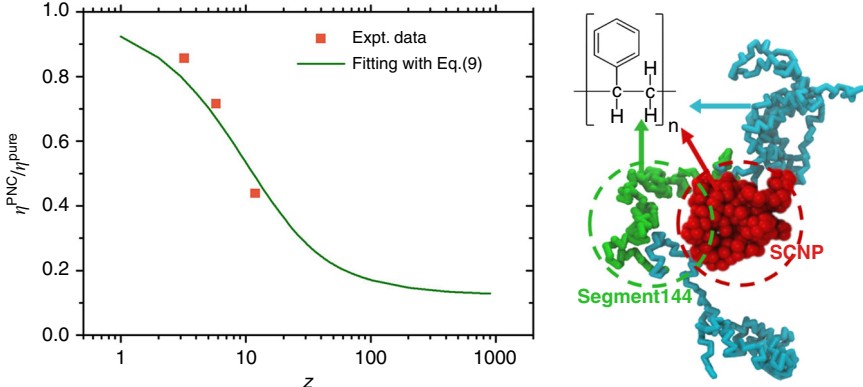

**Fig. 8** Experimentally measured viscosity ratio between composites and pure PS melt, $\eta^{PNC}/\eta^{pure}$, plotted as a function of $Z$. The green line indicates a least squares fitting of these data points using Eq. (9). On the right is a schematic depiction of the length scale of the interactions between SCNP (in red) with polymer chain. The entanglement blob which has a similar size with SCNP is plotted in green.

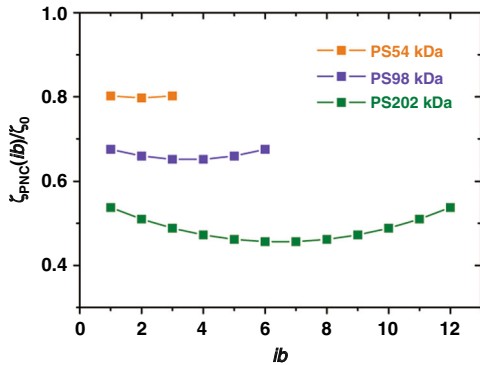

**Fig. 9** Friction coefficient ratios, $\zeta_{PNC}(ib)/\zeta_0$, plotted as a function of the index of entanglement blobs ($ib$) along the chain with the definition in Eq. (3) and the fitting parameters of $\phi\delta = 0.0792$ and $\alpha = 0.0759$ from Fig. 8. $\zeta_{PNC}(Z)$ and $\zeta_0$ are friction coefficient on blobs in the composite and pure PS melt respectively.

reduce the viscosity of the polymer melt, and more importantly such VR effect can be largely amplified in systems with higher polymer molecular weight. Simulation results demonstrate that after adding SCNPs in the system, the extend of the reduction in chain relaxation time is much larger than the reduction in entanglement density. It indicates that the disentanglement effect is not the primary cause for the VR. This is also confirmed by rheological measurements in experiment, which shows that after adding a small amount (2%) of SCNP in polymer melt although there is a large reduction in zero-shear viscosity, the plateau modulus and therefore entanglement density almost does not change.

Based on the results from both simulations and rheology measurements, we propose a mechanism for the abrupt VR and it's $N$-dependence: due to the reduction of the internal degrees of freedom of the soft SCNP, frictions exerted by these SCNPs on surrounding melt polymer chains are reduced. It is important to note that the interaction exerted by SCNP on polymer chains are at a length scale similar with the NP size itself or on the scale of entanglement strand. Such interaction correlates on a scale of ~13 such segmental blobs along the chain backbone. Such correlation along the chain backbone is responsible for the $N$-dependence eventually found in VR effect. A theoretical model is proposed based on these findings. With this model, the viscosity ratio between PNC system, with a small NP loading, and the pure polymer system measured in experiment can be reasonably

described. We hope that our results cannot only shed light on the understanding of VR in the polymer/NP composites and provide new insights for the development of polymer theory for composite systems, but also can be helpful for the relevant material design.

## Data availability

The data that support the findings of this study, including the Supplementary Information, are available from the corresponding author on request.

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

## Acknowledgements

This work is supported by the National Natural Science Foundation of China (21873040, 21522401, 21204029, 21534004, and 91630201). H.J.Q. and Z.Y.L. are also thankful for the support from JLUSTIRT program at Jilin Univeristy. H.J.Q. would like to thank Professor An-Chang Shi at McMaster University, Professor Masao Doi at Beihang University, and Professor Hao-Jun Liang at University of Science and Technology of China for fruitful discussions.

## Author contributions

H.J.Q. conceived the project and designed the both simulation and experiment. T.C. carried out the simulations, part of the rheology measurements and data analysis. Huan-Yu Zhao carried out experiments of synthesis and characterizations. R.S. derived the theoretical model and participated in the analysis of the simulation data. W.F.L., X.X.Z., Y.K.L., and Z.Y.S. carried out part of the rheology measurements. X.M.J. participated in the analysis of the simulation data. T.C. and H.J.Q. co-wrote the manuscript with the input from other co-authors. H.J.Q. and Z.Y.L. co-supervised the project. All authors discussed the results and commented on the manuscript.

## Competing interests

The authors declare no competing interests.
