## [Peer Review File · Nature Communications]

Reviewers' comments:

Reviewer #1 (Remarks to the Author):

The paper proposes a new explanation based on molecular simulations for reported observations of viscosity reduction in entangled polymer melts upon the addition of small nanoparticles comprised of internally crosslinked chains of the same monomer as the linear chains. The nanoparticles are smaller than the tube diameter or entanglement spacing of the linear polymer. For example, in the case of polystyrene, the tube diameter is 8.5 nm while the polystyrene nanoparticle diameter is smaller than 3 nm and hence the nanoparticles diffuse faster than the linear chains. They find that the Rouse modes are accelerated by the presence of small nanoparticles to a greater extent when the linear chains are longer. Hence the time for disentanglement or the tube disengagement time is reduced by the interaction between the small nanoparticles and the linear chains. This is a significant and novel conclusion that the segmental relaxation times are reduced by the interaction with the nanoparticles.

Next, the authors propose that this phenomenon may be attributed to the proximity of nanoparticles to linear chain segments and collisions between them that reduce the monomeric friction coefficient. This leads to faster disengagement of the linear polymer chain from the tube. They hypothesize two different schemes for describing the collision frequency and arrive at a critical degree of polymerization for the linear chains above which the Rouse mode acceleration of will be independent of chain length. This argument is less convincing because the previous sections showed that the segmental relaxation times are reduced by the nanoparticles.

This paper is certainly very interesting and though provoking and will influence thinking in this area.

Reviewer #2 (Remarks to the Author):

This manuscript presents a combined computational and experimental investigation on the viscosity reduction (VR) effect in nanocomposites formed by linear polystyrene (PS) chains and internally-crossed linked single-chain PS nanoparticles (NP) by using molecular dynamics simulations and rheological measurements. It was found that for a fixed NP loading, the VR effect is amplified with the increase of the matrix polymer chain length and a modification to the tube model was introduced to explain this observation. The simulation and experimental finding may provide useful information for further investigation of VR effects in polymer nanocomposites and the paper is clearly written. But the interpretation of the underlying physical mechanism and the proposed model are still phenomenological without microscopic support from the molecular level simulations. For example, there was no direct calculation of the friction reduction in the systems studied by MD simulations. To achieve similar acceleration magnitude in chain relaxation, the NP loading in the MD systems needs to be much higher than that in the experimental systems, indicating that the physical origins of the VR effects found in experiments are not fully captured by the simulations, at least not by the analysis reported in this work. The manuscript thus does not contain sufficient novel and in depth contributions to the research field for being published in Nature Communications. It is more suitable for some subject specific journals.

Below are some detailed comments to the manuscript:

Page 1, left column, "seminar" should be "seminal"

The longest PS chains used in the MD simulations only have around 4 entanglements per chain. It means that most of the simulations were done in no or barely entangled systems. For such short chain lengths, the mean-squared displacement (MSD), $g_1(t)$, of the innermost monomers of the

polymer chains could not demonstrate a true $t^{1/4}$ power law due to the strong contour length fluctuation and constraint release effects. This can already been seen in Figure 1 and could be made more clear by plotting $g_1(t)/t^{1/4}$ against time t . Therefore the claim made in first paragraph of Sec. II B that the pure reptation behavior of $g_1(t) \sim t^{1/4}$ is observed and also the statement that "All of these results indicate that chains in the system are well entangled" could be misleading. This will also affect the applicability of the tube models to such weakly entangled systems.

In Figure 2, the MSDs of the NPs seem to follow similar Rouse-like time dependent behavior as those of the center-of-mass (CMs) of the PS chains. Some explanation needs to be given, such as the dependence on the internal cross-linking density (?).

The correlation functions of the Rouse modes are said to be normalized by $\langle X_p \rangle$ in Sec. II C and also equations (A2, A3), but in Figure 3 they are shown to be normalized by $\langle X_p^2 \rangle$. This inconsistency needs be clarified.

The use of the stretched exponential function to fit the Rouse mode correlation functions is an empirical approach for describing the data without a clear physical meaning of the parameter β_p , especially considering that for entangled chains the modes at small p (i. e., for longer chain segments) are not Rouse. It is then confusing to use the effective relaxation times, as defined in equation (A4), for discussing the acceleration of chain relaxations as shown in Figure 4 and related discussions in Sec. II D. Owing to the different β_p values obtained in the pure melt and the PNCs for the given modes, the ratio between $\tau_{\text{eff}}(p)$ and $\tau_0^{\text{eff}}(p)$ may not be able to reflect the actual acceleration rates in the PNCs.

In Figure 4, the relaxation of the short chain segments (smaller N/p) in PNCs are shown to be slower than those in pure melt. This seems to contradict the picture given later in the paper that the collision of polymer chains with the NPs, typically at length scales equal or smaller than the NP size, can lead to reduction in the friction and so accelerate the chain segment relaxation.

If the PS chains have been fully relaxed in the simulations, as stated in Sec. II C based on the data in Figure 3, it would be more reasonable to calculate the diffusion coefficients of the chains from the $g_3(t)$ data, which should provide more convincing measurement of the accelerated relaxation in PNCs than the effective relaxation times in Figure 4.

Figure 5, it is improper to link data points obtained from different resources and using different NP loading and molecular weights by solid line.

Page 4, since the experiments and MD simulations are measuring different physical quantities and using different PS molecular weights and NP loading, it is not convincing to state that "all of our simulation results are well captured by these experimental measurements."

Page 5, left column, it was stated that "According to tube model, a simple relation $\langle Z_{\text{kink}} \rangle / \langle Z_0^{\text{kink}} \rangle = \tau_d / \tau_0^d$ can be obtained ...". This is not clear to see, considering that for well entangled polymers $\tau_d \sim Z_{\text{kink}}^3$. Some explanations are needed here. Furthermore, since the chain lengths studied in Figure 6 are relatively short (3~4 entanglements), tube model based on pure reptation picture does not apply.

How does the ratio value between the terminal relaxation times, τ_d / τ_0^d , as measured in Figure 7 compare with that between the zero-shear viscosities, η / η_c , presented in Figure 5 for the given PNC sample? This could at least provide some reference for understanding how the results in Figure 4 could be related to viscosity reduction in the simulated systems.

Page 6, the discussions about the collision between NP and polymer chain and the resulted reduction in friction are somewhat confusing. Considering that the NPs are soft particles formed by

internally cross-linked single PS chains, there could many contact points between a NP with a neighboring PS chain due to thermal fluctuations and the collisions between monomers from these two neighbors should take place continuously. How could one physically define a single collision event between a NP and a PS chain?

It is also confusing to state that the NPs are sparsely distributed in the space at the condition that the polymer chains are not long enough. Should the judgement on whether the NPs are sparsely distributed or not depend on the volume fraction of the NPs, rather than the lengths of the PS chains?

If the collision probability is considered to be proportional to the probability of finding a NP in the pervaded volume of a given polymer chain, should this probability be proportional to the product of the pervaded volume of the chain and the volume fraction of the NPs? Why in Figure 8 the number of collisions per unit time is independent of the loading (volume fractions) of the NPs? The dependence of the collision probability on the NP volume fraction should also be included in the theoretical calculations.

It is mentioned on page 6 that "an effective collision between NP and polymer chain is counted only if the center-of-mass of an entanglement strand is located in a distance of 2nm from the NP CM". Such calculation is basically counting the collision events along the confining tube of the chain. If so, the number of collisions during a certain time should be proportional to the volume of the tube region around the chain, giving rise to a different power law dependence on N rather than the $N^{1.5}$ scaling estimated using the pervaded volume of the chain. This will then affect all the theoretical calculations in the remaining part of the paper, not only for the relatively short chain systems but also for the long chain systems.

The original works for the expressions of curvilinear diffusion coefficient and the reptation time with contour length fluctuation effects as used in equations (1-3) should be cited properly.

What are the values of the reduction fractions in friction, δ , used in fitting the data in Figures 9 and 10? Are these values physically reasonable?

Reviewer #3 (Remarks to the Author):

Authors study the relaxation behavior and diffusivity of polymer chains in a polymer-polymer nanocomposite via coarse-grained computer simulation in an attempt to shed some light on the known viscosity reduction in such nanocomposites, and in particular on the effect of molecular weight. In addition, they present experimental results. They argue that disentanglement effects alone are not responsible for the viscosity reduction within the loading regimes studied, and propose a modified reptation theory. The study is certainly of interest for readers of Nat. Commun. The manuscript is well written in most of its parts, and the results scientifically sound, while the theory part is less exciting. Major efforts have been undertaken to study sufficiently long times during the simulations. However, there are some minor and major aspects of this manuscript that are not convincing, and must be improved. The manuscript requires English polishing.

MAJOR

- end of section E: "In short, all of our simulation results are well captured by the experimental measurements". This is misleading. While the experimental result is a viscosity, the viscosity has not been measured during the simulation!! The gap has to be filled.

- The proposed expression (2) for the diffusion coefficient has the major flaw, that it gets negative at large molecular weights. This is healed later by another expression for large N, which is the classical expression. The newly proposed eq (2) is thus only relevant for short chains, but without

knowing anything about delta, it seems quite useless. The quantity delta requires more discussion. The value for delta used here is not even mentioned!

- The comparison in Figs. 9 and 10 with eq (2) and (3) is completely unclear. First of all, ALL fitting parameters have to be mentioned in the captions. Second, the figures should also show the best fit if the refinement ($\delta=0$) is not used. It is also unclear if only eq (3) had been used, or also the expression for large N.

- Appendix A contains too few details about the simulation. Mass of a bead, interaction potentials, choice of time step should be mentioned, so that a reader can redo the calculations and make sense of the dimensional units in all plots.

MINOR

- Authors claim that the large reduction in viscosity can not be simply explained by disentanglement effect. On the other hand they propose the viscosity to be mainly determined by the disentanglement time.

- Page 2: "... with large-scale coarse-grain molecular dynamics simulations". Mention here, and also at the beginning of Section II the appendix A.

- Where is the NP loading in eq (3) hidden? In delta?

- Page 1. "It shows that [the] presence of NPs has almost none effect on the chain dimensions". This had already been reported/discussed in DOI: 10.1080/15583724.2015.1090450

- Caption Fig. 2. "Since the MSD of NP250 does not depend on its loading, ...". I would expect that it starts to change at huge loadings. Please clarify, or give a reason why it doesn't depend on its loading.

- Caption Fig. 6: add $p=1$ to the effective relaxation time. and mention its relationship with τ_d . Actually, the reduced τ_d is shown. The brown circles look like red circles.

- Section F: "The relaxation time ratio is found to be $\tau_d/\tau_{d0} = 88\%$, while Z remains unchanged. Please note that τ_d must not be proportional to Z, as τ_d can also increase due to confinement effects, at least according to Ref. [19]. It should be somewhere stated that you assume, or do not assume, that τ_d is proportional to Z. Further, you do not find the increase of τ_d with increasing NP loading, as observed in the same Ref. Please add a comment.

- Before eq (1) you introduce $X(N)$ proportional to $N^{3/2}$. To derive eq (1) $X(N) = N^{3/2}$ is used. Clean this up.

- eq (2) is for τ_{rep} , Should τ_{rep} be replaced by τ_d here? If not, please explain.

- τ_e in eq (3) should be defined right after eq (3)

- explain the difference between τ_e and primed τ_e'

- Is $X(N) = C N$ proposed here for large N? As how I read the sentences. Later, $X(N) = \delta C N$ has apparently been used, but why?

- "we will have a critical chain length of N^* ". Please define N^* , it is not defined by the expression in the text.

Response to the comments of Referee 1

Referee's Comments:

The paper proposes a new explanation based on molecular simulations for reported observations of viscosity reduction in entangled polymer melts upon the addition of small nanoparticles comprised of internally crosslinked chains of the same monomer as the linear chains. The nanoparticles are smaller than the tube diameter or entanglement spacing of the linear polymer. For example, in the case of polystyrene, the tube diameter is 8.5 nm while the polystyrene nanoparticle diameter is smaller than 3 nm and hence the nanoparticles diffuse faster than the linear chains. They find that the Rouse modes are accelerated by the presence of small nanoparticles to a greater extent when the linear chains are longer. Hence the time for disentanglement or the tube disengagement time is reduced by the interaction between the small nanoparticles and the linear chains. This is a significant and novel conclusion that the segmental relaxation times are reduced by the interaction with the nanoparticles.

Next, the authors propose that this phenomenon may be attributed to the proximity of nanoparticles to linear chain segments and collisions between them that reduce the monomeric friction coefficient. This leads to faster disengagement of the linear polymer chain from the tube. They hypothesize two different schemes for describing the collision frequency and arrive at a critical degree of polymerization for the linear chains above which the Rouse mode acceleration of will be independent of chain length. This argument is less convincing because the previous sections showed that the segmental relaxation times are reduced by the nanoparticles.

This paper is certainly very interesting and though provoking and will influence thinking in this area.

Authors' Response: Many thanks for the positive comments. Since all three referees raised major concerns on the model we proposed. We have been very carefully re-thinking the molecular mechanism for the N -dependent acceleration in chain dynamics we observed. After a thorough discussions between co-authors, we derive a new model in the revised manuscript and we believe this new model is more rigorous than the previous one. For details, Please refer to the derivations in the revised manuscript. They can be found in the section of Molecular Mechanism, on pages 6 and 7.

Secondly, we know from the construction of the Normal Mode analysis, the Rouse Modes defined by Eq. A1 in the APPENDIX A can be used to describe the dynamics of linear polymer chains only when all monomers on polymer chain experience homogeneous frictions. However, in our derivation, the interactions between SCNP and polymer chain decays exponentially on the chain backbone, namely it has a correlation length of α along the chain backbone. Therefore the reduction in friction on monomers are inhomogeneous, which will bring inhomogeneity in frictions for chain monomers along the chain backbone. As a result, we believe Rouse Modes defined by Eq. A1 is

no longer appropriate, even in an effective manner, to describe the relaxation dynamics of short segments at relatively large mode p indices. With the above consideration, we simply drop off data points at small N/p (< 50) values in Fig. 4a in the revised manuscript (which is Fig. 4 in the original submitted version), where we had $\tau^{\text{eff}}/\tau^{\text{eff}}(\phi=0) > 1$.

Response to the comments of Referee 2

Referee's Comments:

This manuscript presents a combined computational and experimental investigation on the viscosity reduction (VR) effect in nanocomposites formed by linear polystyrene (PS) chains and internally-crossed linked single-chain PS nanoparticles (NP) by using molecular dynamics simulations and rheological measurements. It was found that for a fixed NP loading, the VR effect is amplified with the increase of the matrix polymer chain length and a modification to the tube model was introduced to explain this observation. The simulation and experimental finding may provide useful information for further investigation of VR effects in polymer nanocomposites and the paper is clearly written. **But the interpretation of the underlying physical mechanism and the proposed model are still phenomenological without microscopic support from the molecular level simulations.** For example, there was no direct calculation of the friction reduction in the systems studied by MD simulations. To achieve similar acceleration magnitude in chain relaxation, the NP loading in the MD systems needs to be much higher than that in the experimental systems, indicating that the physical origins of the VR effects found in experiments are not fully captured by the simulations, at least not by the analysis reported in this work. The manuscript thus does not contain sufficient novel and in depth contributions to the research field for being published in Nature Communications. It is more suitable for some subject specific journals.

Authors' Response: We find these comments by Referee 2 very insightful and useful. After considering these important suggestions and revising the MS accordingly, we feel that the discussions in our manuscript are much clearer. More importantly, during the revision process, we realized that a better theoretical model can be derived. Details can be found in our following responses to detailed comments. Therefore, we would like to express our heartfelt thanks to referee 2 for his thorough reading of our manuscript and of course for his insightful suggestions.

Below are some detailed comments to the manuscript:

Comment 1: Page 1, left column, “seminar” should be “seminal”

Authors' Response: Thanks for the careful reading, this has been corrected.

Comment 2: The longest PS chains used in the MD simulations only have around 4 entanglements per chain. It means that most of the simulations were done in no or barely entangled systems. For such short chain lengths, the mean-squared displacement (MSD), $g_1(t)$, of the innermost monomers of the polymer chains could not demonstrate a true $t^{1/4}$ power law due to the strong contour length fluctuation and constraint release effects. This can already be seen in Figure 1 and could be made more clear by plotting $g_1(t)/t^{1/4}$ against time t . Therefore the claim made in first paragraph of Sec. II B that the pure reptation behavior of $g_1(t) \sim t^{1/4}$ is observed and also the statement that “All of these results indicate that chains in the system are well entangled” could be

misleading. This will also affect the applicability of the tube models to such weakly entangled systems.

Authors' Response: Thanks for the comment. We carefully revised the corresponding discussions and statement. In addition, we also show the $g_1(t)$ of the innermost monomers for PS250, although the slope in the intermediate regime is even larger than $1/4$, there is a definite decrease of the slope after $t^{1/2}$ regime. We have revised the manuscript (MS) accordingly as follows:

“These results indicate that PS720 chains are entangled. Note that the slope in the intermediate regime is a little larger than a true $t^{1/4}$ power since the system only has ~ 4 entanglements per chain.”

“In addition, $g_1(t)$ is also calculated for PS250 and shown in Figure 1. Although we know that entanglement in this system is very weak since each chain has $N/N_e < 2$ entanglements, a definite decrease of the slope in the intermediate regime is observed after τ_e . Before τ_e , it has exactly the same behaviour as in PS720 system.”

Please find above revisions at the beginning of page 3.

Comment 3: In Figure 2, the MSDs of the NPs seem to follow similar Rouse-like time dependent behavior as those of the center-of-mass (CMs) of the PS chains. Some explanation needs to be given, such as the dependence on the internal cross-linking density (?).

Authors' Response: Thanks for the nice comment. Actually, in our previous studies (Chem. Phys. Lett, 2017, 687, 96-100, cited as ref. 28 in the MS, and JCP, 2016, 145, 106101, cited as ref. 37 in the revised MS), we have investigated the dynamics property of such NP in detail. Where we found that single-chain NP interacts directly with matrix polymers at a length scale of its own size, namely with the chain segment blob which has a similar size as NP. Specifically, such segment blob has 144 monomers and therefore a similar size with the entanglement strand. Therefore we attribute such similarities in NP MSD as those of chains in Fig. 2 to such length-scale dependent interaction between NP and polymer chains. Accordingly, we have revised the MS carefully as follows:

“It is interesting to see that SCNPs seem to follow similar Rouse-like time dependent behavior as those of the CMs of the PS chains, i.e., there is a transition from a sub-diffusive behaviour at short time scale to a Fickian regime at long time scale, though this transition happens a little earlier for SCNPs. We attribute this similarity to the length-scale dependent interaction between these SCNPs and the matrix polymer chains. Specifically, in our previous simulations, [28,37] SCNPs are found to interact directly with chain segment blobs which have similar size as these SCNPs, and we observe that such chain segment blobs have ~ 144 monomers each and a little smaller size than entanglement strand. Therefore the transition from sub-diffusive to Fickian occurs a

little earlier for SCNPs.” It can be found at the end of the 1st paragraph in section II.C.

Comment 4: The correlation functions of the Rouse modes are said to be normalized by $\langle X_p \rangle$ in Sec. II C and also equations (A2, A3), but in Figure 3 they are shown to be normalized by $\langle X_p^2 \rangle$. This inconsistency needs to be clarified.

Authors’ Response: Thanks for careful reading, it’s a typo in the text. The form in the figure is correct. This has been corrected in the revised MS.

Comment 5: The use of the stretched exponential function to fit the Rouse mode correlation functions is an empirical approach for describing the data without a clear physical meaning of the parameter β_p , especially considering that for entangled chains the modes at small p (i. e., for longer chain segments) are not Rouse. It is then confusing to use the effective relaxation times, as defined in equation (A4), for discussing the acceleration of chain relaxations as shown in Figure 4 and related discussions in Sec. II D. Owing to the different β_p values obtained in the pure melt and the PNCs for the given modes, the ratio between $\tau_{\text{eff}}(p)$ and $\tau_{\text{eff}}^0(p)$ may not be able to reflect the actual acceleration rates in the PNCs.

Authors’ Response: Thanks for the professional comment. In the revised MS, we also calculate the relaxation time from the time-dependent auto correlation function of end-to-end vectors, $\langle R(t)R(0) \rangle$, the results are shown in Fig. 4b. We observe that the two algorithms not only offer the same trend, i.e., longer chains will have larger acceleration with the same NP loading, but also the relaxation time ratios are very quantitatively similar to each other. Therefore we add the following in the corresponding discussion as follows:

“Note that this phenomenon still holds if we calculate the effective relaxation time from $\langle R(t)R(0) \rangle$ (see Eqs. A5 and A6 in Appendices), the results are shown in Figure 4(b). Encouragingly, for the same SCNP loading of 20%, relaxation time ratios are found very similar for two algorithms, i.e., values for the last points of each system in Figure 4(a) with $p = 1$ are very similar to the values in Figure 4(b) (triangles).” Please find it on page 4 in the revised MS.

Comment 6: In Figure 4, the relaxation of the short chain segments (smaller N/p) in PNCs are shown to be slower than those in pure melt. This seems to contradict the picture given later in the paper that the collision of polymer chains with the NPs, typically at length scales equal or smaller than the NP size, can lead to reduction in the friction and so accelerate the chain segment relaxation.

Authors’ Response: Thanks for pointing out this important point. This is related to the new theoretical model we proposed in the revised MS. Since all three referees raised major concerns on the model we proposed. We have been very carefully re-thinking the

molecular mechanism for the N -dependent acceleration in chain dynamics we observed. After a thorough discussions between co-authors, we derive a new model in the revised MS and we believe this new model is more rigorous than the previous one. For details, Please refer to the derivations in the revised manuscript. They can be found in the section of Molecular Mechanism, on pages 6 and 7.

Secondly, we know from the construction of Normal Mode analysis, the Rouse Modes defined by Eq. A1 in the APPENDIX A can be used to describe the dynamics of linear polymer chains only when all monomers on polymer chain experience homogeneous frictions. However, in our derivation, the interactions between SCNP and polymer chain decays exponentially on the chain backbone, namely it has a correlation length of α along the chain backbone. Therefore the reduction in friction on monomers are inhomogeneous, which will bring inhomogeneity in frictions for chain monomers along the chain backbone. As a result, we believe Rouse Modes defined by Eq. A1 is no longer appropriate, even in an effective manner, to describe the relaxation dynamics of short segments at relatively large mode p indices. With the above consideration, we simply drop off data points at small N/p (< 50) values in Fig. 4a in the revised manuscript (which is Fig. 4 in the original submitted version), where we had $\tau^{\text{eff}}/\tau^{\text{eff}}(\phi=0) > 1$.

Comment 7: If the PS chains have been fully relaxed in the simulations, as stated in Sec. II C based on the data in Figure 3, it would be more reasonable to calculate the diffusion coefficients of the chains from the $g_3(t)$ data, which should provide more convincing measurement of the accelerated relaxation in PNCs than the effective relaxation times in Figure 4.

Authors' Response: Thanks for the comment. Although the simulations are long enough to equilibrate the system, the longest simulations are on the scale of 20 μs for PS720 systems, as listed in Table A1, we find it is still not enough to get a reliable diffusion constant for the CM of the chains. Especially for the PS720 systems, where polymer chains are very long and their relaxation time in pure melt are on the order of $\sim 3.7 \mu\text{s}$. We find that the Fickian regime in $g_3(t)$ can be used to calculate the diffusion constant is rather short to get a reliable D constant, as can be seen in Figure 1 for pure PS720 melt system.

Comment 8: Figure 5, it is improper to link data points obtained from different resources and using different NP loading and molecular weights by solid line.

Authors' Response: It has been changed to a dash line in the revised MS.

Comment 9: Page 4, since the experiments and MD simulations are measuring different physical quantities and using different PS molecular weights and NP loading, it is not convincing to state that “all of our simulation results are well captured by these experimental measurements.”

Authors' Response: Thanks for the comment, this statement has been removed.

Comment 10: Page 5, left column, it was stated that “According to tube model, a simple relation $\langle Z_{\text{kink}} \rangle / \langle Z^0_{\text{kink}} \rangle = \tau_d / \tau^0_d$ can be obtained ...”. This is not clear to see, considering that for well entangled polymers $\tau_d \sim Z_{\text{kink}}^3$. Some explanations are needed here. Furthermore, since the chain lengths studied in Figure 6 are relatively short (3~4 entanglements), tube model based on pure reptation picture does not apply.

Authors’ Response: Thanks for the comment. Although as pointed out by Referee, the simulated chains in Figure 6 are relatively short (only 3~4 entanglements), we expect that the longest chain PS720 we have in simulation would follow the prediction from the tube model, at least on a qualitative level. However, as we can find in Figure 6, the shorter PS500 systems follows the tube model, while PS700 systems have a large deviation. Important information from this analysis is that disentanglement effect is not the main reason for the N -dependence observed in the acceleration in chain dynamics. For the derivation of the relation, it is based on the fact that the chain relaxation time can be expressed as $\tau_{\text{eff}} \sim \tau_R Z$, τ_R is the Rouse time of the chain, and $Z = N/N_e$ is the entanglement density. In order to make this relation clearer, careful revisions are made in the revised MS, please find them in the right column on page 5, and in the left column on page 6.

Comment 11: How does the ratio value between the terminal relaxation times, τ_d / τ^0_d , as measured in Figure 7 compare with that between the zero-shear viscosities, η / η_c , presented in Figure 5 for the given PNC sample? This could at least provide some reference for understanding how the results in Figure 4 could be related to viscosity reduction in the simulated systems.

Authors’ Response: Thanks for the important suggestion. After we make the comparison, we found a very good agreement between simulation and experiment. This comparison is made in Figure 5 in the revised manuscript. Corresponding discussion are add as following:

“Encouragingly, if we plot the effective terminal relaxation time ratio in the same figure, triangles in Figure5, we find there is a good agreement between simulation and experiment.” Please find it in the 2nd paragraph in section II.E, on page 5, in the left column.

Comment 12: The original works for the expressions of curvilinear diffusion coefficient and the reptation time with contour length fluctuation effects as used in equations (1-3) should be cited properly.

Authors’ Response: They are cited from the book of Doi and Edwards, it is now properly cited as ref.45 in the revised MS.

Comments 13-17 related to the theoretical model:

Comment 13: Page 6, the discussions about the collision between NP and polymer chain and the resulted reduction in friction are somewhat confusing. Considering that the NPs are soft particles formed by internally cross-linked single PS chains, there could

many contact points between a NP with a neighboring PS chain due to thermal fluctuations and the collisions between monomers from these two neighbors should take place continuously. How could one physically define a single collision event between a NP and a PS chain?

Comment 14: It is also confusing to state that the NPs are sparsely distributed in the space at the condition that the polymer chains are not long enough. Should the judgement on whether the NPs are sparsely distributed or not depend on the volume fraction of the NPs, rather than the lengths of the PS chains?

Comment 15: If the collision probability is considered to be proportional to the probability of finding a NP in the pervaded volume of a given polymer chain, should this probability be proportional to the product of the pervaded volume of the chain and the volume fraction of the NPs? Why in Figure 8 the number of collisions per unit time is independent of the loading (volume fractions) of the NPs? The dependence of the collision probability on the NP volume fraction should also be included in the theoretical calculations.

Comment 16: It is mentioned on page 6 that “an effective collision between NP and polymer chain is counted only if the center-of-mass of an entanglement strand is located in a distance of 2nm from the NP CM”. Such calculation is basically counting the collision events along the confining tube of the chain. If so, the number of collisions during a certain time should be proportional to the volume of the tube region around the chain, giving rise to a different power law dependence on N rather than the $N^{1.5}$ scaling estimated using the pervaded volume of the chain. This will then affect all the theoretical calculations in the remaining part of the paper, not only for the relatively short chain systems but also for the long chain systems.

Comment 17: What are the values of the reduction fractions in friction, δ , used in fitting the data in Figures 9 and 10? Are these values physically reasonable?

Authors' Response to comments 13-17: Since we have presented new derivations and a brand-new model, therefore we do not respond to the above comments. Section II. G for the theoretical model is entirely rewritten. Please refer to the corresponding section on pages 6- 8 in the revised MS.

Response to the comments of Referee 3

Referee's Comments:

Authors study the relaxation behavior and diffusivity of polymer chains in a polymer-polymer nanocomposite via coarse-grained computer simulation in an attempt to shed some light on the known viscosity reduction in such nanocomposites, and in particular on the effect of molecular weight. In addition, they present experimental results. They argue that disentanglement effects alone are not responsible for the viscosity reduction within the loading regimes studied, and propose a modified reptation theory. The study is certainly of interest for readers of Nat. Commun. The manuscript is well written in most of its parts, and the results scientifically sound, while the theory part is less exciting. Major efforts have been undertaken to study sufficiently long times during the simulations. However, there are some minor and major aspects of this manuscript that are not convincing, and must be improved. The manuscript requires English polishing.

Authors' Response: We would like to express our heartfelt thanks to referee 3 for his thorough reading of our manuscript and for his very useful suggestions. First of all, we have derived a new model in the revised manuscript (MS), which we believe is more rigorous than the previous one. Details can be found in the revised MS on page 6, and 7 in section II.G. During the revision process, we have also done a very careful language polishing according to Referee.

MAJOR

Comment 1: end of section E: "In short, all of our simulation results are well captured by the experimental measurements". This is misleading. While the experimental result is a viscosity, the viscosity has not been measured during the simulation!! The gap has to be filled.

Authors' Response: Thanks for the comment. Such statements are removed from the MS. In addition, we add some comparisons between simulation and experiment in the revised MS. An example can be found in Figure 5 in the revised MS, where we plot the ratio between terminal relaxation times of composite system against that of the pure polymer melt from simulation, and the viscosity ratio from experiment, we found a very good agreement between simulation and experiment. Although experimental viscosity is different from the terminal relaxation time from simulation, ratios presented in this figure both represent acceleration rate one can achieve after adding SCNPs in the polymer melt.

Comment 2: The proposed expression (2) for the diffusion coefficient has the major flaw, that it gets negative at large molecular weights. This is healed later by another expression for large N , which is the classical expression. The newly proposed eq (2) is thus only relevant for short chains, but without knowing anything about δ , it seems quite useless. The quantity δ requires more discussion. The value for δ used here is not even mentioned!

Comment 3: The comparison in Figs. 9 and 10 with eq (2) and (3) is completely unclear. First of all, ALL fitting parameters have to be mentioned in the captions. Second, the figures should also show the best fit if the refinement ($\delta=0$) is not used. It is also unclear if only eq (3) had been used, or also the expression for large N.

Authors' Responses to Comment 3 and 4: Sorry for the misleading of the previous model we proposed, which is rather phenomenological. In the revised MS, we have derived a new model, which we believe is more rigorous. Please find it on pages 6 and 7 in section II.G.

Comment 4: Appendix A contains too few details about the simulation. Mass of a bead, interaction potentials, choice of time step should be mentioned, so that a reader can redo the calculations and make sense of the dimensional units in all plots.

Authors' Response: Thanks for the comment, these information has been provided in the revised MS.

MINOR

Comment 5: Authors claim that the large reduction in viscosity can not be simply explained by disentanglement effect. On the other hand they propose the viscosity to be mainly determined by the disentanglement time.

Authors' Response: Sorry for the misleading, we have switched it to relaxation time in the revised MS instead of calling it disentanglement time.

Comment 6: Page 2: "... with large-scale coarse-grain molecular dynamics simulations". Mention here, and also at the beginnig of Section II the appendix A.

Authors' Response: Thanks for careful reading, this has been corrected in the revised MS.

Comment 7: Where is the NP loading in eq (3) hidden? In delta?

Authors' Response: As we mentioned above, the theoretical model has been re-derived in the revised MS and all the fitting parameters have been explicitly discussed. Please see the corresponding derivation of the model and the discussions in section II.G in the revised MS on pages 6-8.

Comment 8: Page 1. "It shows that [the] presence of NPs has almost none effect on the chain dimensions". This had already been reported/discussed in DOI: 10.1080/15583724.2015.1090450

Authors' Response: Thanks a lot for reminding us this very important review article. [the] has been added and the reference has been properly cited as follows:

“The same results had been reported in other simulations, a nice review/discussion on this topic can be found in a recent review article by Kröger and coworkers. [29]” Please find it on page, at the bottom of the left column.

Comment 9: Caption Fig. 2. "Since the MSD of NP250 does not depend on its loading, ...". I would expect that it starts to change at huge loadings. Please clarify, or give a reason why it doesn't depend on its loading.

Authors' Response: Sorry for that, it is actually a wrong statement, therefore it is removed from the MS.

Comment 10: Caption Fig. 6: add $p=1$ to the effective relaxation time. and mention its relationship with τ_d . Actually, the reduced τ_d is shown. The brown circles look like red circles.

Authors' Response: It has been added in the figure caption. Sorry for the confusion in colors, instead, we use symbols to differentiate the systems. We are also sorry for the confusion between relaxation time and disentanglement time. It is actually terminal relaxation time at $p = 1$. As we mentioned in the response to **comment 5**, all the symbol τ_d has been replaced by τ^{eff} throughout the revised MS.

Comment 11: Section F: "The relaxation time ratio is found to be $\tau_d/\tau_{d0} = 88\%$, while Z remains unchanged. Please note that τ_d must not be proportional to Z , as τ_d can also increase due to confinement effects, at least according to Ref. [19]. It should be somewhere stated that you assume, or do not assume, that τ_d is proportional to Z . Further, you do not find the increase of τ_d with increasing NP loading, as observed in the same Ref. Please add a comment.

Authors' Response: Thanks for the comment. Accordingly, we have carefully added following comment at the end of the corresponding discussion paragraph.

“Here we have to note that according to Li and coworkers [19], adding solid NPs can cause confinement effects at large NP loadings. We do not observe such confinement effects even at very high loadings of SCNPs, this can be attributed to the inherent softness of these SCNPs.” Please find it on page 6 in the revised MS.

Comment 12: Before eq (1) you introduce $X(N)$ proportional to $N^{(3/2)}$. To derive eq (1) $X(N) = N^{(3/2)}$ is used. Clean this up.

Comment 13: eq (2) is for τ_{rep} , Should τ_{rep} be replaced by τ_d here? If not, please explain.

Comment 14: τ_e in eq (3) should be defined right after eq (3)

Comment 15: explain the difference between τ_e and primed τ_e

Comment 16: Is $X(N) = C N$ proposed here for large N ? hats how I read the sentences. Later, $X(N) = \delta C N$ has apparently been used, but why?

Comment 17: "we will have a critical chain length of N^* ". Please define N^* , it is not defined by the expression in the text.

Authors' Response to comments 12-17: The above comments are all related to our model derivation, however we have presented a bran-new model in the corresponding section and the content related to these comments are all removed. Therefore we are not answering these comments.

Reviewers' comments:

Reviewer #1 (Remarks to the Author):

The authors have responded to reviews by describing a model with varying monomeric friction coefficient along the chain. This involves two adjustable parameters -- a reduction ratio for the friction coefficient and a correlation length along the chain. Fitted values from experiments for the correlation length are equivalent to eight blobs. There appears to be some difficulty still with fitting data for composites based on longer chain linear polymers. The new model is certainly an improvement over arguments given in the first version.

Reviewer #2 (Remarks to the Author):

The authors have made good efforts in addressing the comments raised in the review reports. Most of the questions have been answered reasonably well, but the theoretical model, which is supposed to be the main contribution of this work as indicated by the paper title, remains to be a big concern. Some careful thinking are still needed before the manuscript could be considered for publication, unless some changes are made to avoid the emphasis on the theoretical contributions.

1) The physical meaning of eq. (1) in the newly developed theoretical model is not clear. By definition, the monomer friction coefficient, ζ , is independent of the chain connectivity, as have been used in the Rouse model and also in the current work for the pure melt systems where it is taken to be ζ_0 for all monomers. For a polymer nanocomposite system with a given volume fraction of SCNPs, ϕ , the effective monomer friction, $\zeta(\phi)$, of each individual monomer should be a time- or ensemble-averaged value obtained by averaging over all possible configurations of the system. In other words, these ζ values are determined by taking into account all possible positioning of the SCNPs relative to all chain segments in the system. Following the definition of the monomer friction, one would expect that all monomers in a given composite system should take the same ensemble-averaged $\zeta(\phi)$ value, although their diffusion behaviors (mean-squared displacements) would depend on their chemical positions along the chain. The problem here is fundamentally different from the consideration of a time-dependent behavior where, e.g., if a SCNP is located close to one particular segment of a given chain at a certain time t , one wants to find how the effect of this SCNP-chain segment interaction will propagate along the chain.

It is therefore hard to see the physical reasoning of eq.(1) where the variation of the friction coefficient of a given monomer i with respect to the SCNP fraction ϕ is supposed to have a linear form dependence on the friction coefficients of other monomers at the current ϕ . Such type of systems of differential equations is typically used for dynamic systems, but its usage in describing the viscosity reduction effects needs to be justified.

2) The assumption of the exponential correlation relation for A_{ij} also contracts the independent feature of the monomer frictions by definition.

3) The resulted expression in eq.(5) is essentially the Debye function used for the form factor of scattering of ideal chains, which is a static property related to the chain conformation. Its usage for the dynamic property of monomer friction needs to be justified.

4) The physical meaning the large correlation length α , either 1876 or 1100, is also not clear.

5) It is suggestive for the authors to plot the solution of eq.(3) as a function of the monomer index i , and check how the predicted $\zeta_i(\phi)$ values change along the chain and if the obtained

results have clear physical meaning.

6) The English writing of the revised parts needs to be improved. There are some obvious typos, such as "litter" for "little", and grammatical errors.

Reviewer #3 (Remarks to the Author):

Authors have seriously addressed my concerns and recommendations, the revised manuscript has profitted also from changes in response to the recommendations by the other reviewers. The theoretical part has considerably improved. I am in favor publishing this manuscript, as it should be of interest for readers of Nat. Commun. in its present form.

Responses to Referee: 1

Comments to the Author:

The authors have responded to reviews by describing a model with varying monomeric friction coefficient along the chain. This involves two adjustable parameters -- a reduction ratio for the friction coefficient and a correlation length along the chain. Fitted values from experiments for the correlation length are equivalent to eight blobs. There appears to be some difficulty still with fitting data for composites based on longer chain linear polymers. The new model is certainly an improvement over arguments given in the first version.

Authors' response: Thank you very much for the positive comments for the model we proposed, although there are still some difficulties in fitting data for composites with longer polymers. Such difficulty or uncertainty in the fitting procedure actually also comes from the approximations we made during the model formulation. We made such approximations because in the model the viscosity ratio was written as a function of the chain length, N , namely the number of monomers. If we do not do approximations in the model, we will have an $\mathbf{A}:\mathbf{1}_{N \times N}$ matrix (see it in Eq. 2), which is obviously not treatable during the data fitting with the limited number of experimental points.

To solve the above problem, we have reformatted the model in the revised manuscript, based on the unit of blobs since the SCNP interacts with matrix polymers on the length scale of its own size, which is very similar to the entanglement strand (referred to as blobs). The new formulated model is essentially the same as the former version. The new one is based on the blobs, while the former is based on the monomers. With such a reformatted version, approximations are not needed and the fitting of the experimental data are treatable. $\mathbf{A}:\mathbf{1}_{N \times N}$ matrix will turn into an $\mathbf{A}:\mathbf{1}_{Z \times Z}$ matrix, where Z is the number of entanglement strands per chain. A new fitting can be found in the Figure 8 in the revised manuscript. Corresponding text and equations are revised, please find them on pages 7 and 8 in the revised manuscript.

Responses to Referee: 2

Comments to the Author:

The authors have made good efforts in addressing the comments raised in the review reports. Most of the questions have been answered reasonably well, but the theoretical model, which is supposed to be the main contribution of this work as indicated by the paper title, remains to be a big concern. Some careful thinking are still needed before the manuscript could be considered for publication, unless some changes are made to avoid the emphasis on the theoretical contributions.

Authors' response: Thank you for your comment. According to your comment, we have revised the paper title carefully as “**An unexpected N -dependence of the viscosity reduction in all-polymer nanocomposite**”, to avoid the emphasis on the theoretical contributions. In addition, careful revisions are made in revised manuscript. For details please refer to the replies in the following.

Comment 1) The physical meaning of eq. (1) in the newly developed theoretical model is not clear. By definition, the monomer friction coefficient, ζ , is independent of the chain connectivity, as have been used in the Rouse model and also in the current work for the pure melt systems where it is taken to be ζ_0 for all monomers. For a polymer nanocomposite system with a given volume fraction of SCNPs, ϕ , the effective monomer friction, $\zeta(\phi)$, of each individual monomer should be a time- or ensemble-averaged value obtained by averaging over all possible configurations of the system. In other words, these ζ values are determined by taking into account all possible positioning of the SCNPs relative to all chain segments in the system. Following the definition of the monomer friction, one would expect that all monomers in a given composite system should take the same ensemble-averaged $\zeta(\phi)$ value, although their diffusion behaviors (mean-squared displacements) would depend on their chemical positions along the chain. The problem here is fundamentally different from the consideration of a time-dependent behavior where, e.g., if a SCNPs is located close to one particular segment of a given chain at a certain time t , one wants to find how the effect of this SCNPs-chain segment interaction will propagate along the chain.

It is therefore hard to see the physical reasoning of eq.(1) where the variation of the friction coefficient of a given monomer i with respect to the SCNPs fraction ϕ is supposed to have a linear form dependence on the friction coefficients of other monomers at the current ϕ . Such type of systems of differential equations is typically used for dynamic systems, but its usage in describing the viscosity reduction effects needs to be justified.

Authors' response: Thank you for the comment. (i) First of all, we would like to note that as we have discussed in the manuscript, the SCNPs presented in the system interact directly with matrix polymer chains on the length scale of SCNPs size, i.e., SCNPs can interact directly with polymer chain segments which have a similar size with SCNPs. In the current system, such a segment (144 monomers long) is although smaller but very similar to the entanglement strand. We refer these entanglement strands as blobs in the revised manuscript, correspondingly the model and the equations are rewritten in unit of blobs instead of monomers, for details please refer to our response to Referee 1 and the revised manuscript. (ii) Secondly, as we discussed in the main text, such interaction between SCNPs and matrix chain blob i can reduce the friction and therefore accelerate

the dynamics of blob i . We believe such acceleration in the dynamics of blob i will not disappear immediately while the SCNP leaves, instead such an acceleration effect will decay and propagate along the chain contour due to the chemical connectivity between these blobs. Such contribution is effectively counted in Eq.(1), with term $A_{ij} = \exp(-\alpha|j-i|)$ describes such decay or correlation between blobs along the chain and here $1/\alpha$ is the decay length or correlation length between blobs. Based on the above consideration, we simply use a summation in Eq. (1) to take account of the contribution of each j blob on the same chain to the variation of the friction on blob i at a given SCNP fraction ϕ . To make this point clear, a careful revision is made in the revised manuscript as follows:

“From the above analyses, we know that SCNP in the system interacts with melt PS chains on a length scale of SCNP size or entanglement strand. Knowing such a detailed interaction mechanism between SCNPs and free melt polymer chains will be also the key to understand the observed abnormal N -dependence in chain acceleration or friction reduction. The discussions in the following will be focused on the scale of entangled strands, they are also referred as blobs in the following.

As we discussed above, interactions from SCNPs will effectively reduce the frictions and therefore accelerate the dynamics of surrounding blobs. For instance, if a blob i interacts with an SCNP the friction of the blob i will be reduced and therefore its dynamics will be accelerated. Such acceleration effect will not disappear immediately while the SCNP leaves, instead such an acceleration effect will decay and propagate along the chain contour due to the chemical connectivity between these blobs. Such a dynamic propagation process will effectively also influence the dynamics of the connecting blobs along the chain. Namely, blobs on the same chain can be dynamically correlated. Based on the above consideration, we simply use a summation in the following equation to take account of the contribution of each j blob on the same chain to the variation of the friction on blob i at a given SCNP fraction ϕ ,” Please find it on page 7 in the text before Eq. (1).” Please find these revisions on page 7 in the revised manuscript, before Eq. 1.

Comment 2) The assumption of the exponential correlation relation for A_{ij} also contracts the independent feature of the monomer frictions by definition.

Authors’ response: Thank you for the comments. As we discuss in our response to the comment 1, interaction of SCNP on each polymer chain blob will reduce the friction and therefore accelerates the dynamics of the blob. Such acceleration effect will decay but propagate along the chain contour, A_{ij} describes such decay or correlation between blobs along the chain. Therefore we use an exponential form of $A_{ij} = \exp(-\alpha|j-i|)$. To make it clear, we also make a careful revision in the revised manuscript on page 7 as follows:

“ A_{ij} describes the dynamic decay or correlation between blobs along the chain backbone” Please find it in the text after Eq.(2).

And “ $1/\alpha$ is the decay length or correlation length between blobs.” Find it in the text after Eq.(4).

Comment 3) The resulted expression in eq.(5) is essentially the Debye function used for the form factor of scattering of ideal chains, which is a static property related to the

chain conformation. Its usage for the dynamic property of monomer friction needs to be justified.

Authors' response: Thanks for the comment. As we responded to Referee 1, we can slightly modified the model, some approximations in the last version have been dropped off. Therefore Eq.(5) or Debye function does not exist anymore.

Comment 4) The physical meaning the large correlation length α , either 1876 or 1100, is also not clear.

Authors' response: Thank you for the comment. As we have discussed above in the responses to the comments 1 and 2. SCNPs in the system interacts directly with entanglement blobs in the system. A refitting of the experimental data, after dropping unnecessary approximations in the last version, results in a new decay/correlation length of $1/\alpha = 13$ blobs. We believe it is on a reasonable scale since they take an exponential decay as we described above.

Comment 5) It is suggestive for the authors to plot the solution of eq.(3) as a function of the monomer index i , and check how the predicted $\zeta_i(\phi)$ values change along the chain and if the obtained results have clear physical meaning.

Authors' response: Thank you for the comment. Accordingly the plot is shown in Fig. 9 in the revised manuscript on page 8, corresponding discussions are put on the same page. We see that $\zeta_i(\phi)$ values change along the chain and it has an obvious chain length dependence.

Comment 6) The English writing of the revised parts needs to be improved. There are some obvious typos, such as "litter" for "little", and grammatical errors.

Authors' response: Thank you for the careful reading. We have carefully revised the corresponding typos and grammatical errors throughout the text.

REVIEWERS' COMMENTS:

Reviewer #1 (Remarks to the Author):

This paper presents a well-reasoned explanation for the chain length dependent viscosity reduction reported with addition of small volume fractions of single chain nanoparticles into an entangled polymer matrix. The relevant length scale at which nanoparticles are taken to affect chain dynamics by reducing the friction is taken to be the scale of segment blobs, which is close to the chain entanglement length scale. The correlation length between blobs that arises in their description of this phenomenon is fitted to be of the order of 13 blobs. This paper will stimulate further study of modified constraint release in entangled polymer chains in the presence of nanoparticles.

Reviewer #2 (Remarks to the Author):

The authors have reformulated their theoretical model by using the concept of the frictional correlation between the entanglement blobs, instead of that between individual monomers. This makes the theoretical discussions in the manuscript physically more reasonable.

I think that there is still one key step missing in their reasoning of the model, which is to explain how the argument on the propagation of the frictional reduction effect from one blob to the neighboring blobs along the same chain, which is a process occurring in the time domain, can be used to construct Eq.(1) where the independent variable is the SCNP loading, ϕ , rather than the time t . In other words, how can one physically and/or mathematically translate a time-dependent variation to a concentration-dependent variation of the same physical quantity (blob friction). This missing step can later cause challenges about the foundation of their theoretical model. I would strongly suggest the authors manage to fill in this reasoning gap. Anyway, considering that the simulation and experimental observations are generally interesting, I will still recommend the publication of the manuscript, and leave the answer to this theoretical question as the responsibility of the authors to maintain the quality/validity of their work.

Some minor issues:

In Figure 9, should the label 'z' of the x-axis be replaced by 'i', i.e., the index of the entanglement blobs along the chain? The figure caption should also be changed accordingly.

Page 7, close to the bottom of the right column, "quit good" should be "quite good".

Responses to Referee 2:

Comments to the Author:

The authors have reformulated their theoretical model by using the concept of the frictional correlation between the entanglement blobs, instead of that between individual monomers. This makes the theoretical discussions in the manuscript physically more reasonable.

I think that there is still one key step missing in their reasoning of the model, which is to explain how the argument on the propagation of the frictional reduction effect from one blob to the neighboring blobs along the same chain, which is a process occurring in the time domain, can be used to construct Eq.(1) where the independent variable is the SCNP loading, ϕ , rather than the time t . In other words, how can one physically and/or mathematically translate a time-dependent variation to a concentration-dependent variation of the same physical quantity (blob friction). This missing step can later cause challenges about the foundation of their theoretical model. I would strongly suggest the authors manage to fill in this reasoning gap. Anyway, considering that the simulation and experimental observations are generally interesting, I will still recommend the publication of the manuscript, and leave the answer to this theoretical question as the responsibility of the authors to maintain the quality/validity of their work.

Authors' response: Thank you very for your thorough reading of our manuscript and the thoughtful suggestion. In the original description for equation (1), we used the argument on the propagation of the frictional reduction effect from one blob to the neighboring blobs. It might be not clear, especially the meaning of "propagation" is somehow confusing, we are sorry for that.

In fact, each interaction from SCNP on the polymer chain will reduce the friction and therefore accelerate the dynamics of the interacting blob. Collectively, the dynamics of the entire polymer chain can be accelerated since all blobs have the probability to interact with these SCNPs. On the other hand, acceleration effect on any blob will not disappear immediately while the SCNP leaves. Such acceleration in the dynamics of each blob will also effectively accelerate the dynamics of the neighboring blobs along the chain contour due to the chemical connectivity between these blobs. Namely acceleration in the dynamics of blob i can also effectively accelerate the dynamics of neighboring blobs, such contribution is effectively counted in term A_{ij} in equation (1). To make it clear and avoid unnecessary confusions, corresponding descriptions in the main text are modified accordingly.

Some minor issues:

In Figure 9, should the label 'z' of the x-axis be replaced by 'i', i.e., the index of the entanglement blobs along the chain? The figure caption should also be changed accordingly.

Page 7, close to the bottom of the right column, "quit good" should be "quite good".

Authors' response: Thank you very much for your careful reading. Both of them are corrected.